

# Whole-genome analysis of *Escherichia coli* isolated from wild Amur tiger (*Panthera tigris altaica*) and North China leopard (*Panthera pardus japonensis*)

Hongjia Li[1,*], Tianming Lan[1,*], Hao Zhai[2], Mengchao Zhou[1], Denghui Chen[1], Yaxian Lu[1], Lei Han[1], Jinpu Wei[1], Shaochun Zhou[3], Haitao Xu[4], Lihong Tian[1], Guangshun Jiang[1] and Zhijun Hou[1]

[1] College of Wildlife and Protected Area, Northeast Forestry University, Harbin, Heilongjiang, China
[2] Ningxia Forestry Project Management Center, Yinchuan, Ningxia, China
[3] Heilongjiang Research Institute of Wildlife, Harbin, Heilongjiang, China
[4] Heilongjiang Siberian Tiger Park, Harbin, Heilongjiang, China
* These authors contributed equally to this work.

Corresponding authors
Guangshun Jiang, jgshun@126.com
Zhijun Hou, houzhijundb@163.com

## ABSTRACT

**Background:** *Escherichia coli* is an important intestinal flora, of which pathogenic *E. coli* is capable of causing many enteric and extra-intestinal diseases. Antibiotics are essential for the treatment of bacterial infections caused by pathogenic *E. coli*; however, with the widespread use of antibiotics, drug resistance in *E. coli* has become particularly serious, posing a global threat to human, animal, and environmental health. While the drug resistance and pathogenicity of *E. coli* carried by tigers and leopards in captivity have been studied intensively in recent years, there is an extreme lack of information on *E. coli* in these top predators in the wild environment.

**Methods:** Whole genome sequencing data of 32 *E. coli* strains collected from the feces of wild Amur tiger (*Panthera tigris altaica*, *n* = 24) and North China leopard (*Panthera pardus japonensis*, *n* = 8) were analyzed in this article. The multi-locus sequence types, serotypes, virulence and resistance genotypes, plasmid replicon types, and core genomic SNPs phylogeny of these isolates were studied. Additionally, antimicrobial susceptibility testing (AST) was performed on these *E. coli* isolates.

**Results:** Among the *E. coli* isolates studied, 18 different sequence types were identified, with ST939 (21.9%), ST10 (15.6%), and ST3246 (9.4%) being the most prevalent. A total of 111 virulence genes were detected, averaging about 54 virulence genes per sample. They contribute to invasion, adherence, immune evasion, efflux pump, toxin, motility, stress adaption, and other virulence-related functions of *E. coli*. Sixty-eight AMR genes and point mutations were identified. Among the detected resistance genes, those belonging to the efflux pump family were the most abundant. Thirty-two *E. coli* isolates showed the highest rate of resistance to tetracycline (14/32; 43.8%), followed by imipenem (4/32; 12.5%), ciprofloxacin (3/32; 9.4%), doxycycline (2/32; 6.3%), and norfloxacin (1/32; 3.1%).

**Conclusions:** Our results suggest that *E. coli* isolates carried by wild Amur tigers and North China leopards have potential pathogenicity and drug resistance.

## INTRODUCTION

*Escherichia coli* represents a major cause of morbidity and mortality worldwide (*Da Silva & Mendonça, 2012*). Essentially, *E. coli* is sensitive to almost all clinically relevant antimicrobial agents; however, with the widespread use of antibiotics, the resistance rate of *E. coli* to a variety of drugs has gradually increased, and the infections caused by its drug-resistant strains have tended to increase, and its drug-resistance has continued to transform (*Liu et al., 2021*). Over the past decades, an increasing number of resistance genes have been identified in *E. coli* isolates, and these resistance genes have been transferred horizontally enabling *E. coli* to acquire resistance, *e.g.*, through resistance plasmids, or other mobile genetic elements, transposons, and gene cassettes in class I and class II integrons (*Poirel et al., 2018*).

One of the biggest obstacles to control and treat infectious diseases in animals is the resistance of the causative organisms to antibiotics (*Duangurai et al., 2022*). As antibiotic resistance increases in wildlife populations, previous antibiotic treatments are becoming ineffective, and fighting infectious diseases will become increasingly challenging (*Lagerstrom & Hadly, 2021*). While the drug resistance and pathogenicity of *E. coli* carried by tigers and leopards in captivity have been studied intensively in recent years, there is an extreme lack of information on *E. coli* in these top predators in the wild environment. By studying *E. coli* drug resistance in wild tigers and leopards, we can indirectly assess the spread of *E. coli* drug resistance genes in animal microcosms under the selective pressure of antibiotic use by humans, thus reflecting the ecological condition of wild tigers and leopards.

Several medications are used in traditional antimicrobial therapy to target the pathogen's various functions. However, with every new drug and antibiotic used worldwide, bacteria continue to evolve new mechanisms to evade this drug-mediated killing at an alarming rate, a phenomenon known as Antimicrobial Resistance (AMR). Interest in AMR bacteria and AMR genes isolated from wildlife and the environment has recently grown. Several wildlife species have been found to harbor bacteria resistant to antimicrobial (*Costa et al., 2008*; *Dolejska, Cizek & Literak, 2007*; *Poeta et al., 2005*), and wildlife has been identified as a possible source of AMR bacteria and AMR genes (*Arnold, Williams & Bennett, 2016*). Wild Amur tigers and North China leopards, as rare and large wild animals in China, play a pivotal role in maintaining biodiversity and ecological balance within their habitats. However, the emergence of antibiotic-resistant *E. coli* in these endangered wild animals could pose a threat to their health and the ecosystems they inhabit. The antibiotic resistance genes of *E. coli* can spread through water sources, the food chain, or contact with humans and other wildlife (*Koutsoumanis et al., 2021*), leading to other individuals or species in the wildlife population becoming resistant as well, thus exacerbating the problem of antibiotic resistance and increasing the risk of spreading antibiotic-resistant pathogens in humans and other animals. The spread of drug-resistant

*E. coli* in wildlife could disrupt the balance of ecosystems, where potentially pathogenic and resistant organisms from any one of these ecosystems could easily move to another ecosystem (*Collignon & McEwen, 2019*), leading to over-infection of some species, which could have indirect effects on other organisms, affecting multiple links in the ecological chain, including the food chain and natural interactions, and thus potentially posing threats.

In recent years, with the rapid development of genomics, whole genome sequencing technology has shown its unique advantages in species identification, drug resistance, virulence prediction, and genetic evolutionary analysis (*Purushothaman, Meola & Egli, 2022*). The distribution of bacterial resistance and virulence genes discovered through whole genome sequencing is then used to infer potential bacterial drug resistance and virulence phenotypes, which is crucial for the prevention and treatment of bacterial illnesses. Here, we performed whole-genome sequencing and comparative analysis of 32 *E. coli* isolates from the feces of wild Amur tiger and North China leopard.

## MATERIALS AND METHODS

### Bacterial isolation

From 2012 to 2015, twenty-four fecal samples (H1–H24) from the Amur tiger were randomly collected with the permission of the management authorities of The Changbai Mountain Area in Jilin Province and the Wandashan Mountain National Nature Reserve in eastern Heilongjiang Province, including five samples in 2012, nine in 2013, six in 2014 and four in 2015, total 24. In November 2020, eight fecal samples (B1–B8) of North China leopards were randomly collected from the Liupan Mountains in southern Ningxia, China (Table S1). Fecal samples were transported to the laboratory after aseptic collection and stored at −80 °C for the next step in the experiment. The sampler did not introduce any harmful substances when collecting the feces, which would not disturb the animal's habitat.

### Isolation and identification of *E. coli* and whole genome sequencing

A sterile cotton swab was inserted into the middle of the fecal sample, dipped into an appropriate amount of feces, and spread evenly on MacConkey agar, and then placed into a 37 °C constant temperature incubator for 12 h, and pink single colonies could be observed. Then use the inoculation loop to inoculate the bacteria onto Eosin Methylene Blue agar until single colonies with black purple color and metallic luster grow, and then inoculate onto Nutrient Agar and incubate at 37 °C for 12 h until white single colonies grow.

The white single colonies were picked and tested using the Tubes for biochemical identification of Enterobacteriaceae (Hopebio, Qingdao, China). Subsequently, the genomic DNA of the strain was extracted using a genomic DNA purification kit (Tiangen, Beijing, China) and amplified by PCR with universal primers for bacterial 16S rRNA gene, and the 16S rRNA gene fragment of the strain was amplified and sent to Jilin Comate Bioscience Ltd., Jilin, China. After confirming the bacteria as *E. coli* based on the

sequencing results and biochemical identification, subsequent experiments were performed.

Before library preparation, the DNA was fragmented and screened, and then the MGIEasy Universal DNA Library Preparation Reagent Kit (MGI, Shenzhen, China) was used for library preparation. In the process of library preparation, the DNA fragments were firstly end-repaired and dA tails were added, followed by junction ligation and purification of ligation products. Next, PCR amplification, PCR product purification, PCR product quality control, denaturation, and single-stranded cyclization were performed, followed by enzymatic digestion, enzymatic digestion product purification, and enzymatic digestion product quality control. The digested products were then sequenced on an MGISEQ-2000 gene sequencer from BGI Shenzhen, China, at 150 bp bipartite.

### Antimicrobial susceptibility testing

Antimicrobial susceptibility testing was performed using the WHO-recommended Kirby-Bauer drug-sensitive paper diffusion method according to the standards of the Clinical and Laboratory Standards Institute (CLSI) (*Díez-Aguilar et al., 2015*). We selected five drug-sensitive papers from three classes, including fluoroquinolone antibiotics (ciprofloxacin, norfloxacin), tetracycline antibiotics (tetracycline, doxycycline), and penicillin antibiotics (imipenem). The selection of these drugs was based on resistance phenotypes corresponding to predicted resistant genotypes, and antibiotics of medical and veterinary concern were also considered.

### Reads processing, assembly, and annotation

To ensure the reliability of subsequent credit analysis results, the quality of sequencing raw reads needs to be assessed first, and low-quality data needs to be removed. Here FastQC 0.11.9 (available online at https://www.bioinformatics.babraham.ac.uk/projects/fastqc/, accessed on 2 July 2023) was used to quality control the original sequencing reads. Then the data were filtered using fastp 0.23.2 (*Chen et al., 2018*) to remove adaptors, primers, and low-quality reads. To obtain a series of contigs for subsequent analysis, the sequence splicing of the Clean Data obtained after fasp filtering was performed using SPAdes 3.15.4 (*Bankevich et al., 2012*), where the parameter k-mer size list was set to 55, 65, 75, and 95.

The sequence splicing results obtained from the assembly were then statistically analyzed using Quast 5.2.0 (*Gurevich et al., 2013*) and Busco 5.6.1 (*Seppey, Manni & Zdobnov, 2019*). The analysis aimed to assess the integrity of the genome assembly.

### Bioinformatics analysis

The assembled sequences of *E. coli* isolates were analyzed at the Center for Genomic Epidemiology (available online at https://www.genomicepidemiology.org/services/index. html, accessed on 15 February 2023) to identify multifocal sequence types, serotypes, plasmid replicons, virulence genes, and evolutionary relationships, respectively. Multi-locus sequence typing was performed by running MLST 2.0 (*Larsen et al., 2012*) based on the Achtman scheme for *E. coli*. Serotype identification of assembled sequences by SerotypeFinder 2.0 (*Joensen et al., 2015*) with a selected threshold of 90% identity and

60% total serotype gene length. PlasmidFinder 2.1 (*Carattoli & Hasman, 2020*) was used to study plasmid replicon sequences with thresholds of 90% identity and 60% minimum length, respectively. Abricate 0.5 with the virulence factor database (VFDB) (*Chen et al., 2016*) was used for identifying virulence genes, with parameters set to default values. The whole genome of *E. coli* MG1655 (NC_000913.3) was used as the reference, and assemblies of the 32 strains included in this study were uploaded to the CSI Phylogeny service to construct a phylogeny based on the concatenated alignment of high-quality SNPs. Whole genome data of some ST939 and ST10 strains were downloaded from NCBI for in-depth phylogenetic analysis with sequenced samples from this study. Parsnp 2.0.3 (*Kille et al., 2024*) was used to identify single nucleotide polymorphisms (SNPs) in the *E. coli* genome set and to estimate a phylogenetic tree based on these SNPs. The evolutionary tree was constructed using RAxML 8.2.13 (*Stamatakis, 2014*) with the maximum likelihood method and the bootstrap replication value was set to 1,000.

Assembly files generated by SPAdes 3.15.4 are used as input files to ResFinder 4.1 (*Bortolaia et al., 2020*) and CARD (*Alcock et al., 2023*) to identify both genes and point mutations that confer antimicrobial resistance.

# RESULTS

## Antimicrobial susceptibility testing

In the present study, it was found that among the 32 *E. coli* isolates, there were some differences in the rate of resistance to different drugs through antimicrobial susceptibility testing. Specifically, the highest resistance rate to tetracycline was (14/32; 43.8%), followed by imipenem (4/32; 12.5%), ciprofloxacin (3/32; 9.4%), doxycycline (2/32; 6.3%) and norfloxacin (1/32; 3.1%) (Table 1). Notably, one sample showed resistance to all five antimicrobials (Table S2).

## Genomic characteristics of *Escherichia coli*

Thirty-two strains of *E. coli* isolated from fecal samples of wild Amur tigers (*Panthera tigris altaica*, n = 24) and North China leopards (*Panthera pardus japonensis*, n = 8) were subjected to whole-genome sequencing. The original reads were assembled from scratch to produce overlapping clusters ranging in size from 4.50 Mbp to 5.40 Mbp, with an overall GC content between 50.17% and 50.93%. The number of coding sequences ranged from 4,246 to 5,186, and all samples had a 100% complete BUSCO rate. Consequently, the quality of all assembled overlapping clusters of the 32 *E. coli* isolates was deemed sufficient for genome-wide analysis. The number of coding sequences ranged from 4,246 to 5,186, thus the quality of all assembled overlap clusters of the 32 *E. coli* isolates was considered sufficient for genome-wide analysis. The general characteristics of the *E. coli* genome can be found in Table S3.

## MLST, serotypes and plasmid characterization

Among the *E. coli* isolates studied, 18 different sequence types were identified, with ST939 (21.9%), ST10 (15.6%), and ST3246 (9.4%) being the most prevalent. Other clones were also observed: n = 2 each from ST58, ST126, and singletons from ST11910, ST4038, ST164,

**Table 1 Antimicrobial susceptibility testing results of 32 *Escherichia coli* isolates.**

| Class of drug | Antimicrobial susceptibility testing | | |
|---|---|---|---|
| | **Resistant number** | **Intermediate number** | **Sensitive number** |
| Tetracycline | 14 | 3 | 15 |
| Imipenem | 4 | 10 | 18 |
| Ciprofloxacin | 3 | 3 | 26 |
| Doxycycline | 2 | 2 | 28 |
| Norfloxacin | 1 | 1 | 30 |

**Figure 1 The evolutionary relationship of core genome-SNPs in the 32 *E. coli* isolates from *Panthera pardus fontanierii* and *Panthera tigris altaica*.** The phylogenetic tree was constructed using the core SNPs identified from 32 *E. coli* genome sequences and an *E. coli* reference genome downloaded from NCBI. The analysis was performed using CSI Phylogeny 1.4 (https://cge.food.dtu.dk/services/CSIPhylogeny/) with the Maximum Likelihood method and a default bootstrap replication value of 1,000. The *E. coli* phylogeny indicates (from left to right) sequence type (ST), serotype, isolation date, and plasmid replicon type for each strain.

ST1611, ST2079, ST1377, ST5418, ST7188, ST1378, ST2715, ST1380, ST69 and ST422 (Fig. 1).

*In silico* plasmid replicon typing revealed that the IncF-type (59.4%) plasmid was the most common among the isolates included in this study. Other replicon types included IncX1, IncY, p0111, and IncI2. Among the IncF-type plasmids, IncFII (46.9%) was the most prevalent, followed by IncFIB, IncFIA, and IncFIC (Fig. 1).

We identified multiple serotypes in a collection of 32 *E. coli* isolates. The most predominant serotype was O174:H23 (21.9%), followed by O1:H32 (15.6%). Four isolates

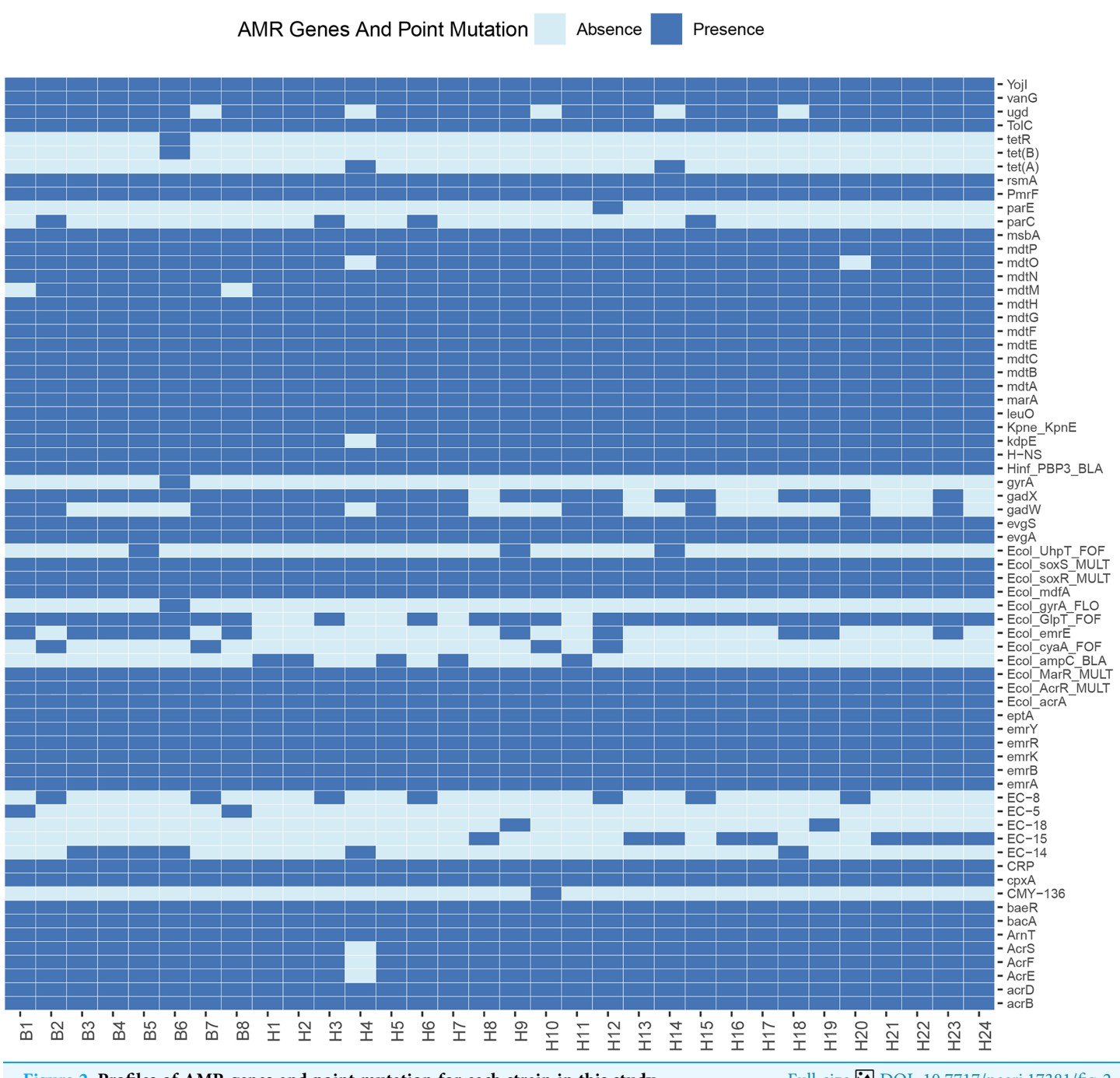

**Figure 2** Profiles of AMR genes and point mutation for each strain in this study.

were founding the serogroups O17/O44/O77:H46. Two isolates were found in the serogroups Ont: H5. The detection rates of serum groups Ont:H14, O88:H21, O99: H21, O93:H28, O20:H8, O125ab:H19, O8:H19, O8:H8, Ont:H11, O74:H52, O16: H48, O117:H14, O2/O50:H25, O9/O158:H18, and O17/O44/O77:H18 were the lowest (Fig. 1).

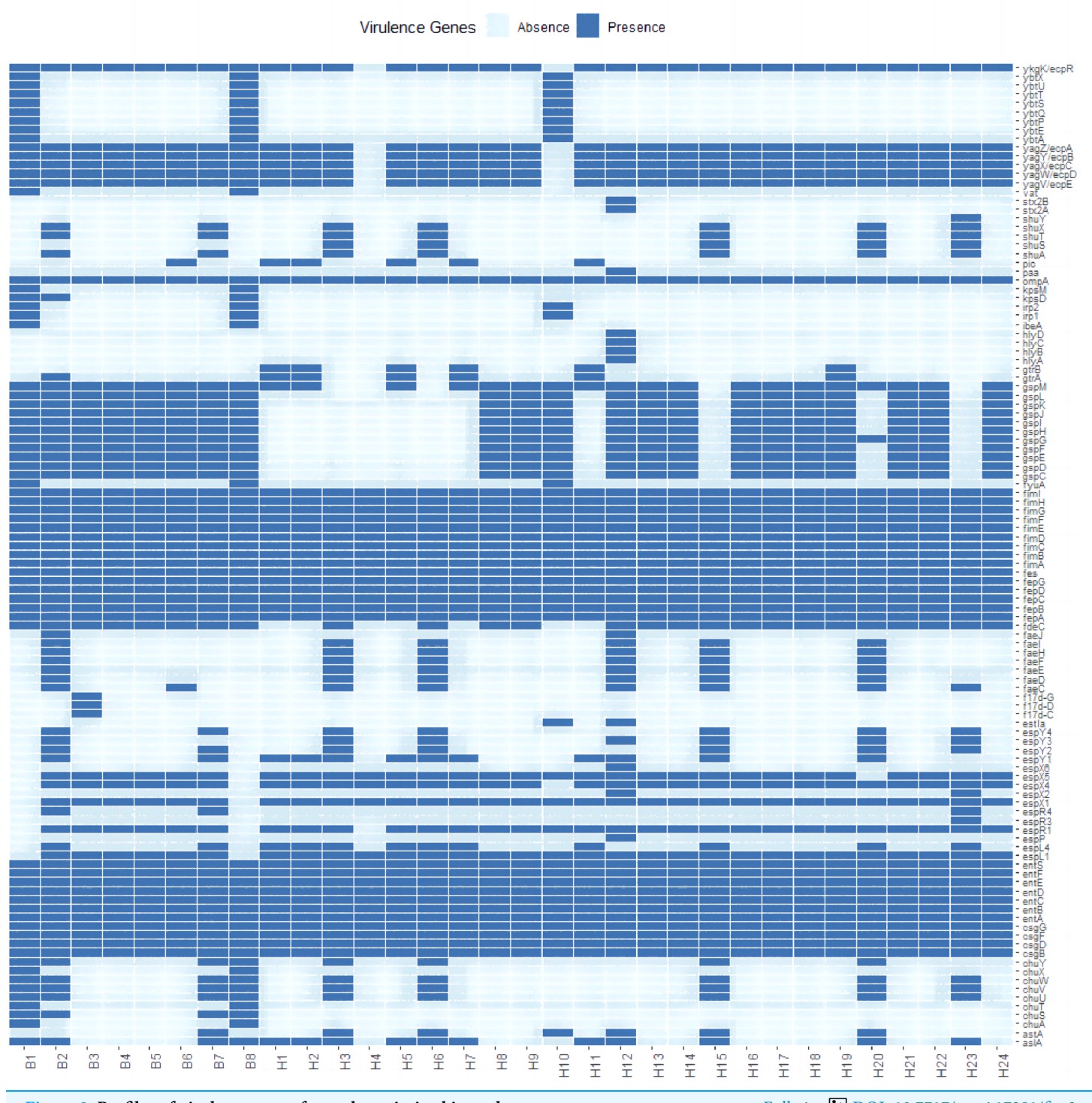

**Figure 3 Profiles of virulence genes for each strain in this study.**

## Distribution of virulence factors and antibiotic-resistance genes

Sixty-eight resistance genes and point mutations were identified. To further understand the differences in resistance correlation between different sources, a presence/absence matrix was drawn to show the distribution of resistance genes and point mutations (Fig. 2).

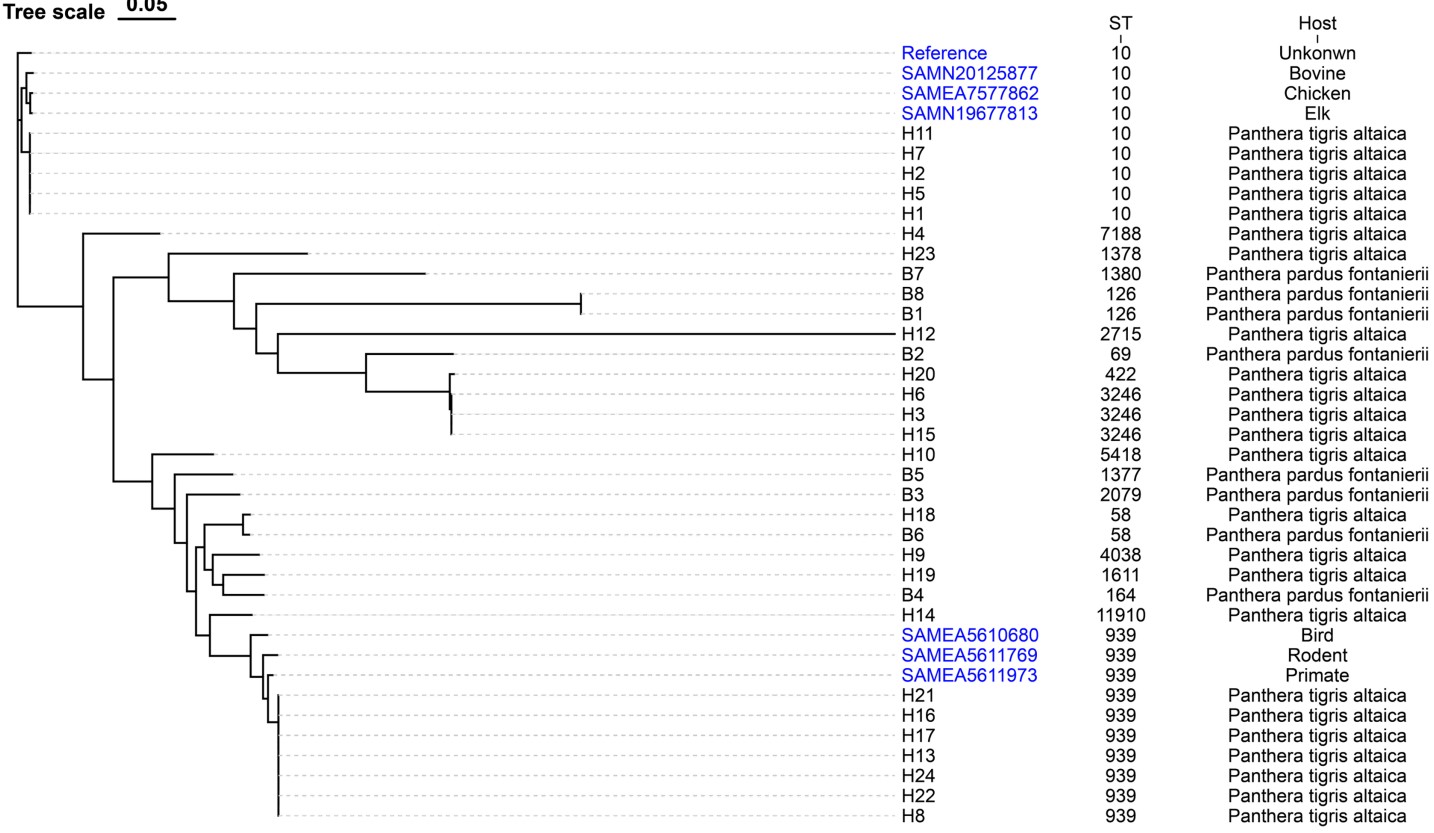

**Figure 4 Phylogenetic tree of *E. coli* strains based on core genome SNPs.** The data used to construct the evolutionary tree include sequenced samples from 32 *E. coli* strains carried by wild Amur tigers and North China leopards, as well as some whole-genome data of *E. coli* with sequence types ST939 and ST10 downloaded from NCBI, with the reference sequence of *E. coli* MG1655. Blue color indicates that the sample was downloaded from NCBI.

Of all the resistance genes and point mutations detected, those encoding fluoroquinolones were the most common, including genes such as *acrB*, *emrA*, *emrB*, *H-NS*, and *rsmA*. This was followed by genes encoding tetracyclines, such as *emrK*, *emrY*, *evgA*, and *evgS*, which were detected in the genomes of all strains.

A total of 111 virulence factors were detected in 32 *E. coli* isolates. To further understand the differences in virulence correlation between different sources, a presence/absence matrix was drawn to show the distribution of virulence genes (Fig. 3). The presence/absence matrix indicates that most strains have similar virulence genes. These virulence genes contribute to invasion, adherence, immune evasion, efflux pump, toxin, motility, stress adaption, and other virulence-related functions of *E. coli*.

## Phylogenetic analysis

The evolutionary relationships of the core genomic SNPs of 32 *E. coli* strains from wild tiger leopard feces are shown in Fig. 1. The maximum SNP distance between each strain was 59,699 SNPs, and the minimum distance was four SNPs. According to the branch lengths, the *E. coli* isolates with serotype O174:H23 (H16, H17, H13, H22, H24, H21, H8) had closer evolutionary distances to each other than to the other branches, with the

number of SNPs differing from 4 to 19 (Table S4). As expected, closely grouped in the generated tree were *E. coli* isolates with the same sequence type and close or identical serotypes (Fig. 1). The phylogenetic tree based on core genomic SNPs was used to further investigate the transmission of strains ST939 and ST10 among different animals. From the evolutionary tree, it can be seen that *E. coli* with sequence types ST939 and ST10 are also present in other animals and clustered in a team with *E. coli* with sequence types ST939 and ST10 carried by wild Amur tigers and North China leopards (Fig. 4).

## DISCUSSION

*E. coli* is widely present in nature, water and food, as well as the respiratory tract and intestinal tract of animals in the normal flora, as the intestinal resident bacteria, in the deterioration of conditions and other pathogenic infections are easy to follow the disease. Some resistance studies have found widespread resistance in *E. coli* from captive tiger sources, and the susceptibility of different pathogenic *E. coli* strains to antibiotics varies widely, including resistance to ampicillin, gentamicin, and chloramphenicol (*Qiu et al., 2016*). Here we performed whole-genome sequencing of 32 strains isolated from wild Amur tiger and North China leopard feces. Among these isolates, we detected 68 resistance genes and point mutations. Of all the resistance genes and point mutations detected, genes encoding fluoroquinolones were the most prevalent, including the genes *acrB*, *emrA*, *emrB*, *H-NS*, and *rsmA*. This was followed by genes encoding tetracyclines, such as *emrK*, *emrY*, *evgA*, and *evgS*, which were detected in the genomes of all strains. Notably, genes from the exocytosis pump family accounted for the largest proportion of the detected resistance genes. This result suggests that the efflux pumps may be associated with antibiotic resistance in the investigated strains. The results of the antimicrobial susceptibility testing showed that among the 32 *E. coli* isolates, there were significant differences in the resistance rates to different drugs. Specifically, the resistance rate to tetracycline was relatively high at 43.8%, but the resistance rates to imipenem, doxycycline, norfloxacin, and ciprofloxacin were relatively low, none of which exceeded 10%. This result suggests that drug-resistant *E. coli* has appeared in the wild populations of the Amur Tigers and the North China Leopards.

In recent years, as more and more virulence factors are understood in *E. coli*, the pathogenic processes of some of these virulence factors have increasingly come into focus. The pathogenicity of *E. coli* is the result of the synergistic effect of multiple virulence factors, and some studies have shown that *E. coli* of tiger origin has a variety of virulence factors, including *Stx2f*, *fimC*, *cnf-1*, *fyuA*, *iroc*, *irp2*, *etc.*, which may be related to the severity of the infection and the course of the disease may also be an important cause of the disease (*Zhang, 2022*). By comparison with the virulence factor database, 111 virulence genes were identified in *E. coli* isolates. The virulence genes with the highest detection rates included the genes *fimA*, *fimB*, *fimC*, *fimD*, *fimE*, *fimF* and *fimH*, which were detected in the genomes of all strains. We also detected the *stx2A* and *stx2B* gene, which produces Shiga toxin, in an *E. coli* isolate. Clinical symptoms of Shiga toxin-producing *E. coli* infections can include hemorrhagic colitis, bloody or watery diarrhea, and potentially fatal

hemolytic uremia (HUS), which can lead to sudden kidney failure. However, it is unclear whether these pathogenic *E. coli* cause disease in these endangered wild Amur tiger and North China leopard, or whether they contribute to the continued decline of populations. It is clear that further research into the microbial diversity of this Amur tiger and North China leopard should be undertaken as part of future investigations to aid their recovery efforts. In this study, whole genome sequencing data showed that 32 *E. coli* isolates were highly serotypically diverse, with a total of 18 different serotypes identified, including four isolates with no O serotypes identified, and those belonging to the same serotype showed extensive similarity and clustered together phylogenetically. The pathogenicity of *E. coli* is linked to the type of O-antigen serotype, with different pathogenic *E. coli* having different dominant O-antigen serotypes. In our study, we identified the dominant serotype O1 of *E. coli* associated with urinary tract infections and the dominant serotype O125ab of enteropathogenic *E. coli*. with serotype O125, first identified in 1952 during a diarrheal outbreak in London (*Taylor & Charter, 1952*) has since been isolated from diarrheal patients worldwide (*Croxen et al., 2013*) and has been recognized as an important pathogenic EPEC serotype by the World Health Organization. However, it is unclear whether these pathogenic serotypes of *E. coli* cause disease in endangered wild Amur tigers and North China leopards, or whether they lead to a sustained population decline. Clearly, research on *E. coli* pathogenicity carried by Amur tigers and North China leopards should be further deepened as part of future investigations.

## CONCLUSIONS

In this study, *E. coli* from wild Amur tigers and North China leopards were analyzed by whole genome sequencing, and we detected multiple drug resistance genes in *E. coli* isolates. We also identified several virulence factors with high pathogenicity as well as serotypes, including the *Stx2A* and *Stx2B* gene that produces Shiga toxin, as well as the dominant serotype O1 of *E. coli* associated with urinary tract infections and the dominant serotype O125ab of intestinal pathogenic *E. coli*. At the same time, through antimicrobial susceptibility testing, we found that some of these *E. coli* isolates were resistant to antibiotics such as tetracycline and imipenem. These would be a potential threat to wild Amur tigers and North China leopards. Our findings, therefore, highlight the importance of surveillance programs, particularly given the potential impact on endangered wildlife and human public health.

### Funding
The project was financially funded by the National Key Research and Development Program of China (2023YFF1305402) and the Heilongjiang Province Key Research and Development Program (GA23A902). The funders had no role in study design, data collection and analysis, decision to publish, or preparation of the manuscript.

## Grant Disclosures

The following grant information was disclosed by the authors:
National Key Research and Development: 2023YFF1305402.
Heilongjiang Province Key Research and Development Program: GA23A902.

## Competing Interests

The authors declare that they have no competing interests.

## Author Contributions

- Hongjia Li performed the experiments, analyzed the data, prepared figures and/or tables, and approved the final draft.
- Tianming Lan performed the experiments, analyzed the data, prepared figures and/or tables, and approved the final draft.
- Hao Zhai analyzed the data, prepared figures and/or tables, and approved the final draft.
- Mengchao Zhou analyzed the data, prepared figures and/or tables, and approved the final draft.
- Denghui Chen performed the experiments, analyzed the data, authored or reviewed drafts of the article, and approved the final draft.
- Yaxian Lu analyzed the data, prepared figures and/or tables, and approved the final draft.
- Lei Han analyzed the data, prepared figures and/or tables, and approved the final draft.
- Jinpu Wei analyzed the data, prepared figures and/or tables, and approved the final draft.
- Shaochun Zhou analyzed the data, prepared figures and/or tables, and approved the final draft.
- Haitao Xu analyzed the data, prepared figures and/or tables, and approved the final draft.
- Lihong Tian analyzed the data, prepared figures and/or tables, and approved the final draft.
- Guangshun Jiang conceived and designed the experiments, authored or reviewed drafts of the article, and approved the final draft.
- Zhijun Hou conceived and designed the experiments, authored or reviewed drafts of the article, and approved the final draft.

## Data Availability

The raw sequencing reads are available at the National Center for Biotechnology Information (NCBI): PRJNA979044.

## Supplemental Information

Supplemental information for this article can be found online at http://dx.doi.org/10.7717/peerj.17381#supplemental-information.

## REFERENCES

Alcock BP, Huynh W, Chalil R, Smith KW, Raphenya AR, Wlodarski MA, Edalatmand A, Petkau A, Syed SA, Tsang KK, Baker SJC, Dave M, McCarthy MC, Mukiri KM, Nasir JA,

# PeerJ

Golbon B, Imtiaz H, Jiang X, Kaur K, Kwong M, Liang ZC, Niu KC, Shan P, Yang JYJ, Gray KL, Hoad GR, Jia B, Bhando T, Carfrae LA, Farha MA, French S, Gordzevich R, Rachwalski K, Tu MM, Bordeleau E, Dooley D, Griffiths E, Zubyk HL, Brown ED, Maguire F, Beiko RG, Hsiao WWL, Brinkman FSL, Van Domselaar G, McArthur AG. 2023. CARD 2023: expanded curation, support for machine learning, and resistome prediction at the comprehensive antibiotic resistance database. *Nucleic Acids Research* **51**:D690–D699 DOI 10.1093/nar/gkac920.

Arnold KE, Williams NJ, Bennett M. 2016. 'Disperse abroad in the land': the role of wildlife in the dissemination of antimicrobial resistance. *Biology Letters* **12(8)**:20160137 DOI 10.1098/rsbl.2016.0137.

Bankevich A, Nurk S, Antipov D, Gurevich AA, Dvorkin M, Kulikov AS, Lesin VM, Nikolenko SI, Pham S, Prjibelski AD, Pyshkin AV, Sirotkin AV, Vyahhi N, Tesler G, Alekseyev MA, Pevzner PA. 2012. SPAdes: a new genome assembly algorithm and its applications to single-cell sequencing. *Journal of Computational Biology* **19(5)**:455–477 DOI 10.1089/cmb.2012.0021.

Bortolaia V, Kaas RS, Ruppe E, Roberts MC, Schwarz S, Cattoir V, Philippon A, Allesoe RL, Rebelo AR, Florensa AF, Fagelhauer L, Chakraborty T, Neumann B, Werner G, Bender JK, Stingl K, Nguyen M, Coppens J, Xavier BB, Malhotra-Kumar S, Westh H, Pinholt M, Anjum MF, Duggett NA, Kempf I, Nykäsenoja S, Olkkola S, Wieczorek K, Amaro A, Clemente L, Mossong J, Losch S, Ragimbeau C, Lund O, Aarestrup FM. 2020. ResFinder 4.0 for predictions of phenotypes from genotypes. *Journal of Antimicrobial Chemotherapy* **75(12)**:3491–3500 DOI 10.1093/jac/dkaa345.

Carattoli A, Hasman H. 2020. PlasmidFinder and in silico pMLST: identification and typing of plasmid replicons in whole-genome sequencing (WGS). *Methods in Molecular Biology* **2075**:285–294 DOI 10.1007/978-1-4939-9877-7.

Chen L, Zheng D, Liu B, Yang J, Jin Q. 2016. VFDB 2016: hierarchical and refined dataset for big data analysis–10 years on. *Nucleic Acids Research* **44(D1)**:D694–D697 DOI 10.1093/nar/gkv1239.

Chen S, Zhou Y, Chen Y, Gu J. 2018. Fastp: an ultra-fast all-in-one FASTQ preprocessor. *Bioinformatics* **34(17)**:i884–i890 DOI 10.1093/bioinformatics/bty560.

Collignon PJ, McEwen SA. 2019. One health-its importance in helping to better control antimicrobial resistance. *Tropical Medicine and Infectious Disease* **4(1)**:22 DOI 10.3390/tropicalmed4010022.

Costa D, Poeta P, Sáenz Y, Vinué L, Coelho AC, Matos M, Rojo-Bezares B, Rodrigues J, Torres C. 2008. Mechanisms of antibiotic resistance in Escherichia coli isolates recovered from wild animals. *Microbial Drug Resistance* **14(1)**:71–77 DOI 10.1089/mdr.2008.0795.

Croxen MA, Law RJ, Scholz R, Keeney KM, Wlodarska M, Finlay BB. 2013. Recent advances in understanding enteric pathogenic Escherichia coli. *Clinical Microbiology Reviews* **26(4)**:822–880 DOI 10.1128/CMR.00022-13.

Da Silva GJ, Mendonça N. 2012. Association between antimicrobial resistance and virulence in Escherichia coli. *Virulence* **3(1)**:18–28 DOI 10.4161/viru.3.1.18382.

Díez-Aguilar M, Morosini MI, López-Cerero L, Pascual Á, Calvo J, Martínez-Martínez L, Marco F, Vila J, Ortega A, Oteo J, Cantón R. 2015. Performance of EUCAST and CLSI approaches for co-amoxiclav susceptibility testing conditions for clinical categorization of a collection of Escherichia coli isolates with characterized resistance phenotypes. *Journal of Antimicrobial Chemotherapy* **70(8)**:2306–2310 DOI 10.1093/jac/dkv088.

**Dolejska M, Cizek A, Literak I. 2007.** High prevalence of antimicrobial-resistant genes and integrons in Escherichia coli isolates from Black-headed Gulls in the Czech Republic. *Journal of Applied Microbiology* **103(1)**:11–19 DOI 10.1111/j.1365-2672.2006.03241.x.

**Duangurai T, Rungruengkitkul A, Kong-Ngoen T, Tunyong W, Kosoltanapiwat N, Adisakwattana P, Vanaporn M, Indrawattana N, Pumirat P. 2022.** Phylogenetic analysis and antibiotic resistance of Escherichia coli isolated from wild and domestic animals at an agricultural land interface area of Salaphra wildlife sanctuary, Thailand. *Veterinary World* **15**:2800–2809 DOI 10.14202/vetworld.2022.2800-2809.

**Gurevich A, Saveliev V, Vyahhi N, Tesler G. 2013.** QUAST: quality assessment tool for genome assemblies. *Bioinformatics* **29(8)**:1072–1075 DOI 10.1093/bioinformatics/btt086.

**Joensen KG, Tetzschner AM, Iguchi A, Aarestrup FM, Scheutz F. 2015.** Rapid and easy in silico serotyping of Escherichia coli isolates by use of whole-genome sequencing data. *Journal of Clinical Microbiology* **53(8)**:2410–2426 DOI 10.1128/JCM.00008-15.

**Kille B, Nute MG, Huang V, Kim E, Phillippy AM, Treangen TJ. 2024.** Parsnp 2.0: scalable core-genome alignment for massive microbial datasets. *bioRxiv* DOI 10.1101/2024.01.30.577458.

**Koutsoumanis K, Allende A, Álvarez-Ordóñez A, Bolton D, Bover-Cid S, Chemaly M, Davies R, De Cesare A, Herman L, Hilbert F, Lindqvist R, Nauta M, Ru G, Simmons M, Skandamis P, Suffredini E, Argüello H, Berendonk T, Cavaco LM, Gaze W, Schmitt H, Topp E, Guerra B, Liébana E, Stella P, Peixe L. 2021.** Role played by the environment in the emergence and spread of antimicrobial resistance (AMR) through the food chain. *EFSA Journal* **19(6)**:e06651 DOI 10.2903/j.efsa.2021.6651.

**Lagerstrom KM, Hadly EA. 2021.** The under-investigated wild side of Escherichia coli: genetic diversity, pathogenicity and antimicrobial resistance in wild animals. *Proceedings of the Royal Society B: Biological Sciences* **288(1948)**:20210399 DOI 10.1098/rspb.2021.0399.

**Larsen MV, Cosentino S, Rasmussen S, Friis C, Hasman H, Marvig RL, Jelsbak L, Sicheritz-Pontén T, Ussery DW, Aarestrup FM, Lund O. 2012.** Multilocus sequence typing of total-genome-sequenced bacteria. *Journal of Clinical Microbiology* **50(4)**:1355–1361 DOI 10.1128/JCM.06094-11.

**Liu C, Wang P, Dai Y, Liu Y, Song Y, Yu L, Feng C, Liu M, Xie Z, Shang Y, Sun S, Wang F. 2021.** Longitudinal monitoring of multidrug resistance in Escherichia coli on broiler chicken fattening farms in Shandong, China. *Poultry Science* **100(3)**:100887 DOI 10.1016/j.psj.2020.11.064.

**Poeta P, Costa D, Sáenz Y, Klibi N, Ruiz-Larrea F, Rodrigues J, Torres C. 2005.** Characterization of antibiotic resistance genes and virulence factors in faecal enterococci of wild animals in Portugal. *Journal of Veterinary Medicine, Series B* **52(9)**:396–402 DOI 10.1111/j.1439-0450.2005.00881.x.

**Poirel L, Madec JY, Lupo A, Schink AK, Kieffer N, Nordmann P, Schwarz S. 2018.** Antimicrobial resistance in Escherichia coli. *Microbiology Spectrum* **6**:4 DOI 10.1128/microbiolspec.arba-0026-2017.

**Purushothaman S, Meola M, Egli A. 2022.** Combination of whole genome sequencing and metagenomics for microbiological diagnostics. *International Journal of Molecular Sciences* **23(17)**:9834 DOI 10.3390/ijms23179834.

**Qiu W, Hu T, Wang W, Yu J, Guo P, Zheng Y, Li G, Fan Q. 2016.** Isolation and identification of lethal pathogens of the Amur tiger (Panthera tigris altaica) and drug sensitivity tests. *Chinese Journal of Veterinary Medicine* **52(10)**:38–39 (in Chinese).

**Seppey M, Manni M, Zdobnov EM. 2019.** BUSCO: assessing genome assembly and annotation completeness. *Methods in Molecular Biology* **1962**:227–245 DOI 10.1007/978-1-4939-9173-0.

**Stamatakis A. 2014.** RAxML version 8: a tool for phylogenetic analysis and post-analysis of large phylogenies. *Bioinformatics* **30(9)**:1312–1313 DOI 10.1093/bioinformatics/btu033.

**Taylor J, Charter RE. 1952.** The isolation of serological types of Bact. coli in two residential nurseries and their relation to infantile gastroenteritis. *The Journal of Pathology and Bacteriology* **64(4)**:715–728 DOI 10.1002/path.1700640405.

**Zhang Y. 2022.** Research progress of escherichia coli in tigers. *Modern Animal Husbandry Science and Technology* **2022(11)**:33–36 (in Chinese) DOI 10.19369/j.cnki.2095-9737.2022.11.008.