# Peer review of "Whole-genome analysis of Escherichia coli isolated from wild Amur tiger (Panthera tigris altaica) and North China leopard (Panthera pardus japonensis)"

_PeerJ, doi:10.7717/peerj.17381_

## Round 0.1 · original submission · Major Revisions

I recommend following the reviewers' comments, especially focussing on the antibiotic resistance profiles of the isolates and validating these results, as both reviewers requested; especially with a focus on the difference between phenotypic resistance and resistance prediction based on genomic data.

**Language Note:** The review process has identified that the English language must be improved. PeerJ can provide language editing services - please contact us at copyediting@peerj.com for pricing (be sure to provide your manuscript number and title). Alternatively, you should make your own arrangements to improve the language quality and provide details in your response letter. – PeerJ Staff

·

Basic reporting

The article needs some improvements regarding the language and sentence structure. It is advised for a fluent English speaker to revise it. For example:
line53: " to enable" enabling
line55-58; this sentence should be deleted as this is the definition of conjugation and it is not only limited to E.coli.
line 59: to control and treat

Experimental design

I couldn't find the section of MICs in the Materials and methods and not in the results as well. MICs should be provided and results explained in order to correlate the phenotypic profile with the genomic one.
line 103: what is biochemically identifying E.coli. Please elaborate.
line 127: its better to search for virulence genes in the VFDB and not Virulence genes as it should provide more genes. Also CARD should be used with Resfinder to help detect genes not found in the CGE database.
line154-159: to assess the genome assembly, authors should use BUSCO as it is considered the standard for assessing genome assemblies.

Validity of the findings

line 161: the five unknown STs: where these ST submitted to the database for assigning novel STs? if not it should be.
line264-275: Most of the detected mutations and genes are chromosomally encoded. If the authors suspect a gene to be on a plasmid, the authors have to do conjugation/transformation assay and confirm the presence of the gene and plasmid replicon through PCR and PBRT; the results presented by the authors dont show any part of the plasmids here.

Additional comments

it would be beneficial for the manuscript to include also another phylogeny to compare this sample with samples that has similar STs from NCBI data base or Enterobacter. This can show whether these strains have similar clones in other species.

·

Basic reporting

In this study, Li et al. investigated the genotypic characteristics of (putative) antimicrobial-resistant Escherichia coli strains, isolated from fecal samples of wild Amur tigers and North China leopards. The authors isolated a total of 32 strains, all of which were subjected to whole-genome sequencing and subsequent analysis. Although I believe that the presence of antimicrobial-resistant E. coli in wildlife is an important topic, the current manuscript needs to be substantially revised to meet the requirements of the journal and to appeal to a wider audience.

The English language needs to be clearly revised. In addition, the authors should properly use common technical terms from the field of antimicrobial resistance research. For example, the authors state “[…] we detected 12 AMR genes and 3 additional point mutant genes, which are resistant to a variety of drugs such as phosphomycin, colistin, quinolones.” (lines 43 & 44) or “The ability of E. coli as a donor to pass drug-resistant genes to other bacteria […]” (lines 55 & 56). However, genes cannot be resistant, but they can confer resistance. Also, the fimH gene is not a type I hair (see line 239), but a type I fimbriae. Please correct these and the many other errors throughout the manuscript.

The included literature references are relevant and could support the authors’ statements, especially in the introduction. However, in my humble opinion, the relationship between the intestinal carriage of antimicrobial-resistant E. coli in wildlife species and infections caused by these pathogens is not well explained (e.g., lines 70–72). Is there any epidemiologic information on infections and deaths of tigers and leopards in the reserve area?

The figures are of good quality and the colors used are easy to distinguish. However, the figure legends are not adequate and the authors should definitely provide more information to the reader. For example, the description of Figure 1 (i.e., the phylogenetic tree) should include the method of tree construction (including the alignment approach) and rooting (I assume it is an unrooted tree). I also recommend including a label with the year of isolation. Also, Figure 2 and Figure 3 seem redundant and should be merged.

Experimental design

Lines 92–101: The authors should provide more information about the different isolates from the different time periods. I would recommend a summary table of all isolated E. coli strains with the basic information (e.g., sequence type, year of isolation, host species etc.). Did all samples contain antimicrobial-resistant E. coli, or was the initial number of fecal samples higher and the authors are reporting only those samples that were positive for antimicrobial-resistant E. coli? In addition, I assume that the fecal samples from different carnivores are not clearly distinguishable. How did the authors ensure the correctness of the specified host species?

Lines 103 & 104: Since the study focuses on the presence of antimicrobial-resistant E. coli in wildlife, the authors should provide more information on the isolation process (the method given in the reference is too general). What selection marker was used?

Lines 106 & 107: The author should provide more details about the library preparation and the sequencing on the MGISEQ-2000 platform. What is the reason for the introduction of the abbreviation NGS, since it will not be used again in the text below?

Lines 133–138: I recommend to perform an in-depth phylogenetic analysis of the ST939 and also ST10 strains. This could provide a better insight into the clonal relationships between these strains and indicate a circulation of certain subclones within the reserve or even between different wildlife species.

Lines 139–142: Why were both AMRFinderPlus and the NCBI database used for the prediction of resistance genes?

Genotypic prediction of resistance genes alone is acceptable, but does not replace phenotypic antimicrobial susceptibility testing (AST). Since AST is a standardized and relatively inexpensive technique, authors should validate genotypic results by phenotypic testing.

Lines 143–152: What was the expected outcome for this analysis?

Validity of the findings

The data for this study are deposited and accessible in a repository.

Lines 161–165: The authors should provide a summary of the distribution of the different phylogenetic E. coli backgrounds in the phylogroups (see: 10.11111/1462-2920.14713).

Line 179: Figures 2 and 3 show a presence/absence matrix, not a heat map. Please adapt the statement.

Line 184: The pmrB gene does not confer resistance to colistin per se, but mutations in this gene have been shown to be associated with colistin resistance. Again, AST is needed to verify this finding.

Lines 202–215: It is not clear to me why the authors performed this analysis. It is well known that E. coli has an open pangenome (e.g., 10.1186/s12915-022-01347-7). This analysis neither supports the results of this study nor adds any new knowledge.

Lines 227–230: acrF encodes a component of an efflux pump complex. Therefore, the presence of this gene solely indicates that efflux pumps could be involved in the antimicrobial resistance of the investigated strains. However, the authors only summarize genotypic results and a conclusion of efflux pump activity are not permissible. Similar is true for the mdtM gene.

Lines 230 & 231: “resistance profiles”, this was not tested by the authors. Please rephrase.

Line 239: “type I hairs”, as stated above, this is the wrong description for FimH.

Line 246–250: Is there evidence that Shiga toxin-producing E. coli can cause the mentioned (fatal) symptoms in Amur tigers and North China leopards? Please provide literature references.

Line 267–275: Short read sequencing does not allow unambiguous resolution of plasmid sequences, and unless the predicted resistance genes happen to be on the same contig as the Inc replicon, the authors’ statement “The results of plasmid replicon analysis indicate that wild Amur tigers and North China leopards have a high potential for the spread of drug resistance” (lines 274 & 275) remains speculative.

---

## Round 0.2 · accepted · Accept

Dear authors, the reviewers recommended to accept your revised manuscript without further revisions. I will follow this evaluation. The editiorial decision is "accept".

·

Basic reporting

no comment

Experimental design

no comment

Validity of the findings

no comment

Additional comments

The manuscript has improved after the revision.

·

Basic reporting

The authors have adequately addressed all my comments and suggestions. I have nothing to add or criticize.

Experimental design

No comment

Validity of the findings

No comment

Additional comments

No comment